# Prediction of pathological complete response to neoadjuvant chemotherapy in locally advanced breast cancer by using a deep learning model with [18]F-FDG PET/CT

**Gülcan Bulut**[1☯]*, **Hasan Ikbal Atilgan**[2☯], **Gökalp Çınarer**[3☯], **Kazım Kılıç**[4☯], **Deniz Yıkar**[5☯], **Tuba Parlar**[6☯]

1 Division of Medical Oncology, International Medicana Hospital, Izmir, Turkey, 2 Department of Nuclear Medicine, Mustafa Kemal University Medical School, Hatay, Turkey, 3 Department of Computer Engineering, Faculty of Engineering and Architecture, Bozok University, Yozgat, Turkey, 4 Department of Computer Programming, Yozgat Vocational High School, Bozok University, Yozgat, Turkey, 5 Division of Nuclear Medicine, Hatay Training and Research Hospital, Hatay, Turkey, 6 Department of Computer Technologies, Mustafa Kemal University, Hatay, Türkiye

☯ These authors contributed equally to this work.
* gulcanbulut07@gmail.com

**Data Availability Statement:** There are ethical restrictions on publicly sharing a de-identified minimal data set due to sensitive patient

## Abstract

### Objectives

The aim of the study is 18F-FDG PET/CT imaging by using deep learning method are predictive for pathological complete response pCR after Neoadjuvant chemotherapy (NAC) in locally advanced breast cancer (LABC).

### Introduction

NAC is the standard treatment for locally advanced breast cancer (LABC). Pathological complete response (pCR) after NAC is considered a good predictor of disease-free survival (DFS) and overall survival (OS).Therefore, there is a need to develop methods that can predict the pCR at the time of diagnosis.

### Methods

This article was designed as a retrospective chart study.For the convolutional neural network model, a total of 355 PET/CT images of 31 patients were used. All patients had primary breast surgery after completing NAC.

### Results

Pathological complete response was obtained in a total of 9 patients. The study results show that our proposed deep convolutional neural networks model achieved a remarkable success with an accuracy of 84.79% to predict pathological complete response.

information. Data is available from the secretary of the Mustafa Kemal University Ethics Committee via email (esedatb@hotmail.com) for researchers who meet the criteria for access to confidential data.

**Funding:** The authors received no specific funding for this work.

## Conclusion

It was concluded that deep learning methods can predict breast cancer treatment.

## 1. Introduction

Breast cancer is the most common form of cancer and the second most common cause of cancer death amongst women. Neoadjuvant chemotherapy (NAC) is the standard treatment for locally advanced breast cancer (LABC) and tumor downstage to achieve breast-conserving surgery [1]. Pathological complete response (pCR) after NAC is considered a good predictive marker for disease-free survival (DFS) and overall survival (OS), particularly in patients with more aggressive subtypes such as triple-negative or HER2-positive breast cancer [2, 3]. Therefore, there are numerous studies examining clinicopathological features that can be used to predict pCR in patients receiving NAC [4]. Clinical tumor size (cT) and tumor grade are other clinicopathological features used to predict pCR [4, 5]. Apart from molecular subtypes, no other biomarker, including Ki 67 value and residual cancer, burden has so far been validated as predictive markers for pCR after NAC [6].

Early detection of response to NAC is important to prevent the toxicity of ineffective chemotherapy [7]. Conventional imaging tools including fluorine 18 ($^{18}$F) fluorodeoxyglucose (FDG) PET/CT and magnetic resonance imaging (MRI) have been used in the evaluation of the responders after NAC [8]. PET/CT provides reliable information not only tumor size changes but also about the evaluation of tumor response [9]. $^{18}$F-FDG PET/CT, which has been widely used in imaging of cancers, shows glucose metabolism of cancer cells. $^{18}$F-FDG PET/CT has been used for staging and restaging of cancers and for assessing therapy response. $^{18}$F-FDG PET/CT can provide early detection of response to chemotherapy in locally advanced and metastatic breast cancers [10]. $^{18}$F-FDG PET/CT is a valuable imaging tool for the early assessment of response to NAC in breast cancer [7]. Some researchers study the role of radiomic features from $^{18}$F-FDG PET/CT images to predict pathological complete response to neoadjuvant chemotherapy in locally advanced breast cancer patients [11]. $^{18}$F-FDG PET/CT can allow early detection of pathological complete response after NAC in triple negative breast cancer [12]. Changes in SUVmax value after NAC is associated with pathological complete response [13].

In recent years, deep neural networks have been used effectively in medical image processing field to classify radiological images [14]. The most popular architecture is the convolutional neural networks (CNNs) model. CNNs model minimizes the data pre-processing and learns features from large amounts of data with using convolutions and pooling functions. Numerous studies have been presented regarding the use of CNNs for diagnostic purposes in radiological imaging of the breast [15–17]. However, few studies have reported that it is used to predict pathological complete response to neoadjuvant chemotherapy using the $^{18}$F-FDG PET/CT images [18, 19]. $^{18}$F-FDG PET/CT has a high sensitivity and specificity when predicting breast cancer metastasis [20]. $^{18}$F-FDG PET/CT images were studied in diagnosis, staging, prediction of prognosis and evaluation of response to therapy using CNNs in clinical field including oncology lung cancer, head and neck cancer, prostate cancer, cervical cancer, and sarcomas [21].

To the best of our knowledge, there have not been any study to predict the complete pathological responses of locally advanced breast cancer to neoadjuvant chemotherapy using residual deep neural network model on $^{18}$F-FDG PET/CT images. The primary aim of this study

was to examine whether the clinicopathological features and $^{18}$F-FDG PET/CT imaging by using deep learning method are predictive for pCR after NAC in locally advanced breast cancer.

## 2. Material and method

### 2.1. Patients

The study included 31 patients with a mean age of 54.26±10.71 years (min: 33, max: 82) as can be observed in Table 1. Medical records of the patients who received NAC after diagnosis of LABC in the Medical Oncology Division of Defne Hospital between 2013 and 2020, were retrospectively evaluated. This study was reviewed by the approval of Mustafa Kemal University Medical School Clinical Research Committee with the decision number 13 on the date 29.11.2019. The participants provided written informed consent. All patients details included in the study were de-identified.

As can be seen in Table 1, one patient had radiological T1, 11 patients T2, 17 patients T3 and two patients T4. Seven patients had radiological N1, 20 patients N2 and four patients N3 lymph node metastasis. The mean SUVmax values of the primary tumor were 12.84±7.56 (min: 4.25, max: 32.24). Nine of 31 patients (29.03%) had complete pathological response, while the remaining 22 patients (70.97%) had residual disease on surgical pathological specimens.

The patients were selectively selected according to inclusion and exclusion criteria. The inclusion criteria for the study were female gender, clinical status of II to III according to the 8th Edition tumor-node-metastasis (TNM) classification of the American Joint Committee on Cancer Staging [22], complete blood count performed prior NAC, availability of postoperative pathology reports after surgical procedure. As part of the NAC regimen, the patients were

**Table 1. Patient characteristics.**

| No. of Patients | 31 |
|---|---|
| Age (years) | |
| Mean±SD (min-max) | 54.26±10.71 (33–82). |
| Radiological T Grade | |
| T1 | 1 |
| T2 | 11 |
| T3 | 17 |
| T4 | 2 |
| Radiological N Grade | |
| N1 | 7 |
| N2 | 20 |
| N3 | 4 |
| SUVmax of Primary Tumor | |
| Mean±SD (min-max) | 12.84±7.56 (4.25–32.24) |
| Tumor Receptor Status | |
| ER/PR +, HER2 - | 17 |
| ER/PR -, HER2- | 3 |
| ER/PR-, HER2+ | 7 |
| ER/PR+HER2 + | 4 |
| Complete Pathological Response to NAC | |
| Present (%) | 9 (29.03%) |
| Absent (%) | 22 (70.97%) |

administered taxane-based regimens (paclitaxel 80 mg/m2 for 12 weeks or 4 cycles of docetaxel 75 mg/m2, every 3 weeks) combined and anthracycline-based regimens (4 cycles of doxorubicin 60 mg/m2 and cyclophosphamide 600 mg/m2, or cyclophosphamide 600 mg/m2 and epirubicin 50 mg/m2, every 3 weeks). Human epidermal growth factor receptor 2 (HER2) positive patients treated with trastuzumab (At the time of the study, there was no access to pertuzumab in neoadjuvant therapy in our country) [23]. The exclusion criteria are absence of [18]F-FDG PET/CT images, incomplete NAC, and absence of surgical treatment.

## 2.2. Subtypes of breast cancer

Tumor size and lymph node involvement level were evaluated in all patients included in the study. Needle biopsy specimens performed before NAC and tissues removed by surgical procedure were subjected to histopathological and immunohistochemical (IHC) examinations.

Estrogen receptor (ER), progesterone receptor (PR), and HER2 status were determined by the IHC method; the specimens of patients with a staining level of $\geq 1\%$ in the tumor cells were considered as having positive ER and PR status; further, HER2 status was regarded as positive if it was 3+ and negative if it was $\leq$1+. Then, HER2 status was confirmed by fluorescence in situ hybridization (FISH) for patients with 2+ HER2 status on IHC testing. Breast cancer was classified into four subtypes: HR+, HER2+; HR+, HER2-; HR-, HER2+; and HR-, HER2-.

## 2.3. pCR

In the postoperative pathological evaluation, the absence of invasive tumors in the breast tissue or lymph node (regardless of the presence of an in-situ component) was defined as pCR (ypT0/ypN0).

## 2.4. [18]F- FDG PET/CT imaging

All patients had one PET/CT imaging in the staging of breast cancer before NAC. Blood glucose level was measured after fasting at least 6 hours. If the blood glucose level was less than 180 mg/dL, 3.7 MBq/kg [18]F-FDG was injected intravenously. Images were acquired from vertex to mid thigh approximately 60 minutes after injection, using an integrated PET/CT scanner (Siemens Biograph mCT, Siemens Healthcare, Erlangen, Germany). CT scans were acquired with tube voltage 120 kV, effective tube current intensity 80–250 mAs, rotation time 0.5 s, detector configuration 16×1.2 mm, slice thickness 5 mm and used for attenuation correction and fusion with PET images. PET scan was acquired with 1.5–2 minute acquisition time for each 6–8 bed positions. PET, CT and PET/CT fusion images were evaluated with SyngoVia workstation (Siemens AG, Muenchen, Germany). Thirty one PET/CT imagings of 31 patients were used for deep learning process. First the PET/CT fusion slices of the primary tumor in the breast was detected for the deep learning process. One out of every three slices were recorded as Digital Imaging and Communication in Medicine (DICOM) format [24]. DICOM format was converted to JPEG format with Syngo FastView software (Siemens AG, Muenchen, Germany) (Fig 1). Deep convolutional neural networks models were applied these images. The sizes of JPEG images were between 57–97 KB and the resolution was 96 dpi with 1605x1064 pixels. The SUVmax values of the primary tumors were calculated on DICOM images with Syngo FastView software by drawing VOIs on tumors and using the following formula: SUVmax = activity concentration / (injected activity/patient weight). SUVmax = tracer uptake in ROI / (injected activity/patient weight).

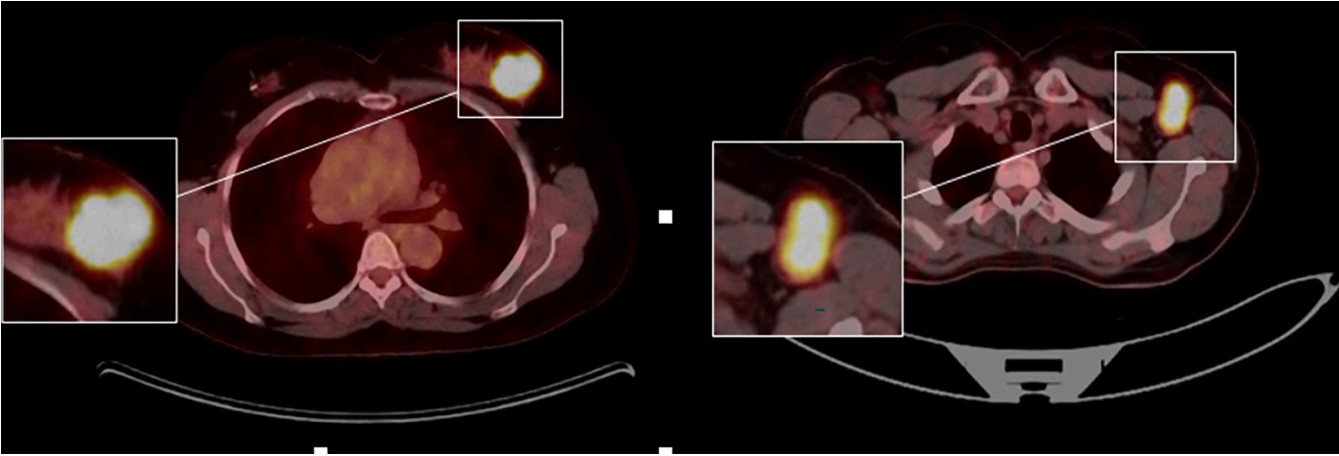

**Fig 1. [Deep convolutional neural networks model]: Diagram of image cropping for the residual deep convolutional neural networks algorithm.** The cubic shaped region-of-interest was selected at the largest cross-sectional area of the lesion and resized to 224 × 224 pixels. (a) pCR: 0 (b) pCR: 1.

The study mainly focused on PET/CT images, deep convolutional neural networks algorithm was used for only the PET/CT fusion images. SUVmax values were noted but not added to the proposed deep learning process.

## 2.5. Model development

Convolutional Neural Networks (CNN) are the most popular type of artificial neural network in the field of image classification [25]. Numerous studies have been published on CNNs with regards to their potential use in diagnosing cancer based on radiological images. Deep neural networks are difficult to train because of the vanishing gradients problem. Residual networks (ResNet) architecture was proposed by He et al. [26] in the field of image recognition to solve this problem.

ResNet architecture presents a residual learning block for reducing the deterioration of deep neural networks. As can be observed in Fig 2, the shortcut connection adds the input $x$ to

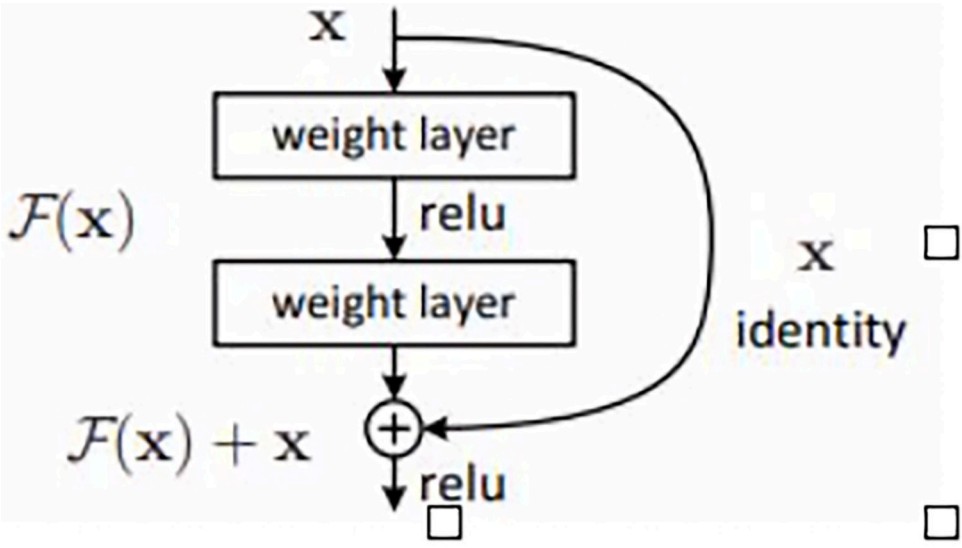

**Fig 2. Residual learning block.**

the output $F(x)$ function. The resulting function $H(x) = F(x)+x$ is transmitted to the next layer and the process is repeated for the other residual learning blocks. In this way, the learning rate, and the number of learned parameters increase with depth. This sometimes negatively affects the network training time and learning speed. For this reason, it is tried to prevent this situation using the bottleneck technique. With the bottleneck technique, the 224x224 input image is reduced to 56x56. ReLu and normalization operations are applied in each layer. Each layer obtains triple convolutional layers with filters. In ResNet, a bulk normalization layer is added after each convolutional layer, which is an important feature that distinguishes it from other architectures. With the normalization process, the learning and error rate is also reduced. The model allows inputs to propagate more quickly between layers using residual learning blocks. By using non-linear ReLu in the model, the neuron density is lowered, and the number of operations is reduced. The ResNet architecture produces better accuracy without increasing the complexity of the model by using a feedforward network with a shortcut connection which adds new inputs into the network and generates new outputs.

ResNet has many variants with different numbers of layers such as 18, 34, 50, 101, and 152. We choose ResNet-152 as it achieves the best accuracy among other ResNet models. Fig 3 illustrates the basic architecture of ResNet-152 for ImageNet [26].

| Layer name | Output size | 152 Layer | |
|---|---|---|---|
| conv1 | 112x112 | 7x7,64, stride2 | |
| conv2.x | 56x56 | 3x3 max pool, stride 2 | |
| | | 1x1, 64<br>3x3, 64<br>1x1, 256 | x3 |
| conv3.x | 28x28 | 1x1, 128<br>3x3, 128<br>1x1, 512 | x8 |
| conv4.x | 14x14 | 1x1, 256<br>3x3, 256<br>1x1, 1024 | x36 |
| conv5.x | 7x7 | 1x1, 512<br>3x3, 512<br>1x1, 2048 | x3 |
| | 1x1 | average pool, 1000d-fc, softmax | |
| FLOPs | | $11.3 \times 10^9$ | |

**Fig 3. The basic architecture of ResNet-152.**

## 3. Results

For the convolutional neural networks model, a total of 355 PET/CT images of 31 patients were used. The images consist of two classes, 107 (30.1%) belong to patients who complete responded to neoadjuvant chemotherapy, and 248 (69.9%) belong to patients who did not complete responded to neoadjuvant chemotherapy. In the experiments, the data set was divided into 80% training and 20% testing. ResNet-152 architecture was used for the training and testing using five-fold cross validation. Experiments were carried out in a cloud environment using the Python programming language on the K80 Tesla graphics card. Fast.ai library is used for hyperparameter selections and network setup.

We performed five-fold cross-validation to improve the reliability of the ResNet-152 model. Sensitivity, specificity and F-score values obtained from the test set for each fold (Table 2). C1, C2, C3, C4, and C5 represent the training-testing groups, respectively. The "# of Samples" column shows the number of images belonging to each class in the relevant group.

As can be observed in Table 2, 87% sensitivity, specificity and F-score values were obtained on the weighted average for the experiment performed on the C1 group. 85% F-score with weighted average 86% sensitivity and specificity in C2 group, 81% sensitivity, 80% specificity, 81% F-Score in C3 group. In the C4 group, the best values of the study, 92% sensitivity, specificity and F-score were obtained. In the C5 group, 80% sensitivity, 79% specificity and 80% F-score values were obtained on a weighted average.

When the results are examined, it is observed that the classification model gives successful results for each training set (Fig 4).

The performance of the classification model was measured at 84% accuracy, 90% AUC score, 85% sensitivity, 84% specificity and 85% F-score when averaged using five-fold cross-validation (Table 3).

ROC curves showing the true positive rate and false positive rate for the C1, C2, C3, C4 and C5 experimental groups are shown in Fig 4. As a result of classification performed on test sets in these experiments, C1, C2, C3, C4, and C5 86%, 89%, 90%, 98% and 89% AUC scores were obtained, respectively. On average, a 90% AUC score was obtained.(Fig 5).

**Table 2. ResNet-152 cross-validation results.**

|  | pCR to NAC | Sensitivity | Specificity | F-Score | # of Samples |
|---|---|---|---|---|---|
| Fold C1 | 0 | 0.90 | 0.92 | 0.91 | 49 |
|  | 1 | 0.80 | 0.76 | 0.78 | 21 |
|  | Weighted Avg | 0.87 | 0.87 | 0.87 | 70 |
| Fold C2 | 0 | 0.85 | 0.96 | 0.90 | 49 |
|  | 1 | 0.87 | 0.62 | 0.72 | 21 |
|  | Weighted Avg | 0.86 | 0.86 | 0.85 | 70 |
| Fold C3 | 0 | 0.88 | 0.84 | 0.86 | 50 |
|  | 1 | 0.65 | 0.71 | 0.68 | 22 |
|  | Weighted Avg | 0.81 | 0.80 | 0.81 | 72 |
| Fold C4 | 0 | 0.96 | 0.92 | 0.94 | 50 |
|  | 1 | 0.83 | 0.91 | 0.87 | 22 |
|  | Weighted Avg | 0.92 | 0.92 | 0.92 | 72 |
| Fold C5 | 0 | 0.87 | 0.82 | 0.85 | 50 |
|  | 1 | 0.64 | 0.73 | 0.68 | 22 |
|  | Weighted Avg | 0.80 | 0.79 | 0.80 | 72 |

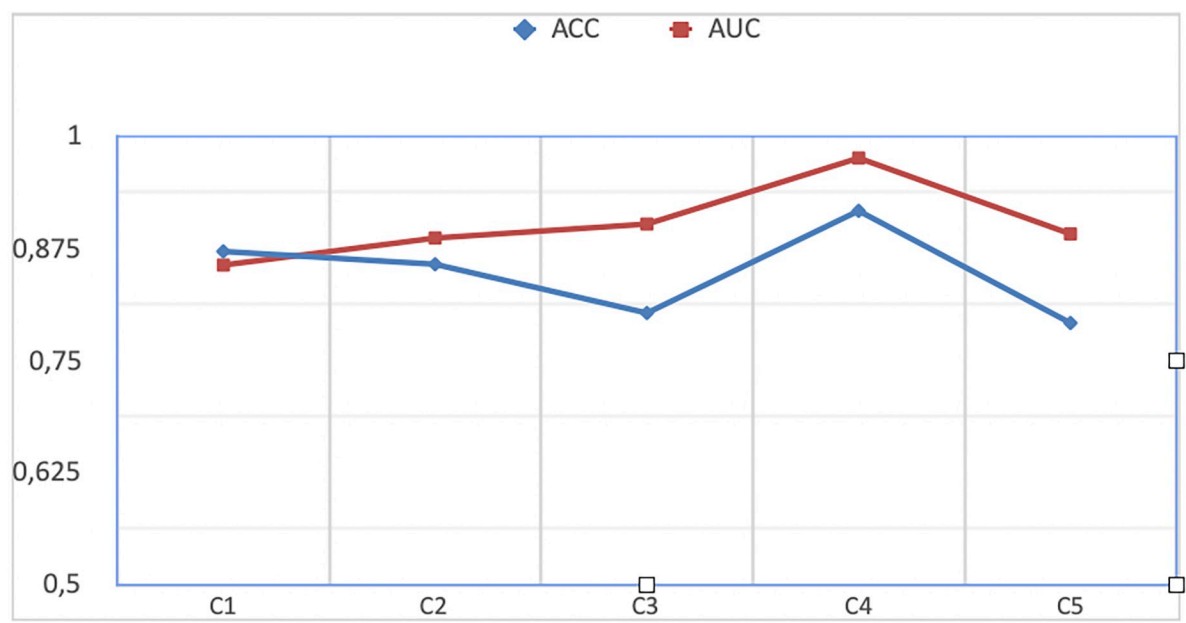

**Fig 4. [Area under the curve]] The classification performance using AUC (area under the curve) and accuracy analysis for each group.**

## 4. Discussion

The experimental results show that our proposed deep neural network model achieved a remarkable success with an accuracy of 85% to predict pathological complete response to neoadjuvant chemotherapy using fused $^{18}$F-FDG PET/CT images. A total of 355 $^{18}$F-FDG PET/CT images were obtained from 31 locally advanced breast cancer patients and the tumor regions were cropped by specialist doctors.

Artificial intelligence has been used in many studies such as diagnosis, staging or response to chemotherapy. The literature review shows that deep learning methods have achieved the significant improvements for medical image analysis. A three-dimensional deep convolutional neural networks (3D DCNN) model used to recognize benign pleural disease from malignant pleural mesothelioma with FDG PET/CT images. They had clinical features with an accuracy of 82.4% and AUC of 0.896 [27].

Antunovic et al. [11] predicted pCR to NAC in breast cancer with AUC values from 0.70 to 0.73 using multiple regression model. Li et al. [28] found an accuracy of 0.857 (AUC = 0.844) on the training split set and 0.767 (AUC = 0.722) on the independent validation set to determine impacts of radiomic features on pCR to NAC using $^{18}$F-FDG PET/CT images in breast cancer. When patient age was combined with PET, the accuracy of training split set increased to 0.857 (AUC = 0.958) and the accuracy of the independent validation set increased to 0.8 (AUC = 0.73). They use unsupervised and supervised machine learning methods to select the most important features for their model. Hwang et al. [29] conducted a study to detect aLN metastasis. They obtained an accuracy of 77.1% in ultrasound images, an accuracy of 77.9% in MRI images, and an accuracy of 81.1% in PET/CT images. Our proposed model outperforms with an accuracy of 85%

**Table 3. The classification results of ResNet-152 model.**

|  | Accuracy | AUC | Sensitivity | Specificity | F-Score |
|---|---|---|---|---|---|
| Average | 0.8479 | 0.9021 | 0.852 | 0.848 | 0.850 |

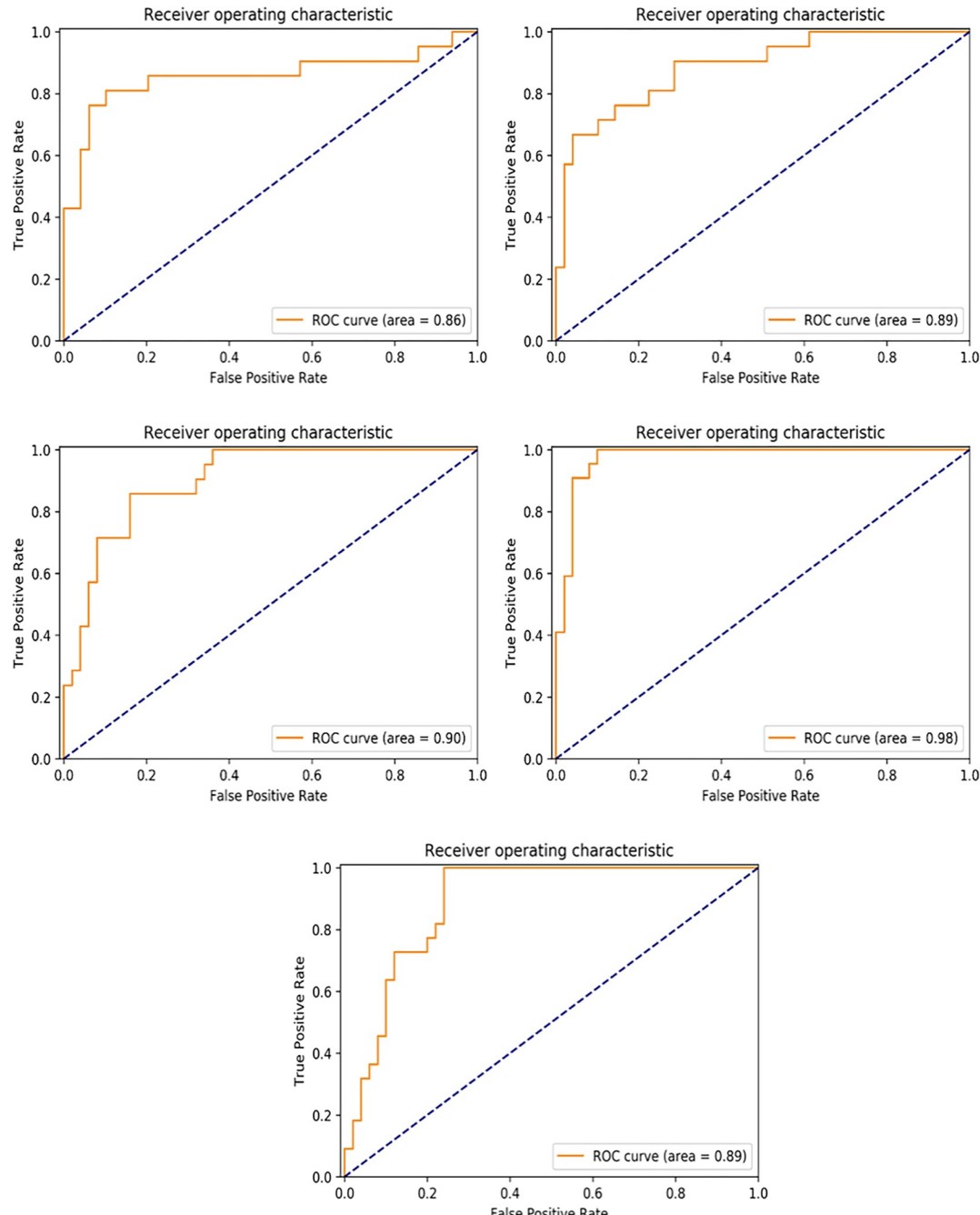

**Fig 5. [Receiver operating characteristic curves] Receiver operating characteristic (ROC) curves of ResNet-152 on $^{18}$F-FDG PET/CT images for each cross-validation group.**

in PET/CT images. Choi et al. examined pCR to NAC using CNNs with PET/CT and PET/MR images in advanced breast cancer patients. When compared with SUVmax values, their model increased the AUC value from 0.652 to 0.886 and accuracy from 84% to 97% in baseline, and AUC value from 0.687 to 0.980 and accuracy from 70% to 95% in interim [8].

Despite the strengths of the current study, some limitations should be taken into account when considering the results. First, as with any retrospective study, there was a bias inherent

in its nature. Although patient selection is from retrospective files, pet images were reprocessed in this study and is a cohort study. Second, power calculation was not done for estimation of sample size selected in the study. All patients who match the inclusion criteria were included in the study. Addition of other indexes such as SUVmax, SUVmean, TLG, MTV, T and N grade, HU would make the analysis better. The last limitation, perhaps the most important limitation, is that the study was conducted with rather small number of patients. Future prospective and large database studies should be used to further validate and investigate our results.

## 5. Conclusions

We examined the pathological complete response to neoadjuvant chemotherapy of locally advanced breast cancer patients using a deep convolutional neural network model on fused [18]F-FDG PET/CT images. Our proposed ResNet-152 deep convolutional neural networks architecture achieved a remarkable classification performance with an accuracy of 84.79%. In conclusion, deep convolutional neural networks can have significant impact to analyze [18]F-FDG PET/CT images.

## Author Contributions

**Data curation:** Hasan Ikbal Atilgan, Deniz Yıkar.

**Formal analysis:** Gökalp Çınarer.

**Investigation:** Kazım Kılıç.

**Validation:** Tuba Parlar.

**Writing – original draft:** Tuba Parlar.

**Writing – review & editing:** Gülcan Bulut.

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
