## [Decision Letter · Decision Letter 0]

14 Feb 2023

PONE-D-22-31819Prediction of pathological complete response to neoadjuvant chemotherapy in locally advanced breast cancer by using a deep learning model with 18 F-FDG PET/CTPLOS ONE

Dear Dr. Bulut,

Thank you for submitting your manuscript to PLOS ONE. After careful consideration, we feel that it has merit but does not fully meet PLOS ONE’s publication criteria as it currently stands. Therefore, we invite you to submit a revised version of the manuscript that addresses the points raised during the review process. Please submit your revised manuscript by Mar 31 2023 11:59PM. If you will need more time than this to complete your revisions, please reply to this message or contact the journal office at plosone@plos.org. Please include the following items when submitting your revised manuscript:A rebuttal letter that responds to each point raised by the academic editor and reviewer(s). You should upload this letter as a separate file labeled 'Response to Reviewers'.A marked-up copy of your manuscript that highlights changes made to the original version. You should upload this as a separate file labeled 'Revised Manuscript with Track Changes'.An unmarked version of your revised paper without tracked changes. You should upload this as a separate file labeled 'Manuscript'.

We look forward to receiving your revised manuscript.

Kind regards,

Erdal Karavaş, M.D.

Academic Editor

PLOS ONE

Journal Requirements

https://www.ejmi.org/pdf/Significance%20of%20NeutrophilLymphocyte%20Ratio%20and%20TrombocyteLymphocyte%20Ratio%20in%20Predicting%20Complete%20Pathological%20Response%20in%20Patients%20with%20Local%20Advanced%20Breast%20Cancer-32029.pdf

https://journals.plos.org/plosone/article?id=10.1371%2Fjournal.pone.0259622

In your revision ensure you cite all your sources (including your own works), and quote or rephrase any duplicated text outside the methods section. Further consideration is dependent on these concerns being addressed.

no funding

NO authors have competing interests

5. Please ensure that you include a title page within your main document. You should list all authors and all affiliations as per our author instructions and clearly indicate the corresponding author.

6. Please ensure that you refer to Figure 5 in your text as, if accepted, production will need this reference to link the reader to the figure.

Additional Editor Comments:

Dear Author;

Thank you for submitting your valuable manuscript to PLOS ONE. We would like to inform you that the peer review process of your manuscript is complete and requires "Major Revision". Although it has several useful points, it needs Major Revision to be reviewed again. To assist you in making your alterations, we are enclosing the comments below.

Kind regards.

Reviewers' comments:

Reviewer's Responses to Questions

**Comments to the Author**

1. Is the manuscript technically sound, and do the data support the conclusions?

Reviewer #1: Partly

Reviewer #2: Yes

2. Has the statistical analysis been performed appropriately and rigorously? 

Reviewer #1: Yes

Reviewer #2: N/A

3. Have the authors made all data underlying the findings in their manuscript fully available?

Reviewer #1: Yes

Reviewer #2: Yes

4. Is the manuscript presented in an intelligible fashion and written in standard English?

Reviewer #1: No

Reviewer #2: Yes

5. Review Comments to the Author

Reviewer #1: The aim of this study was to predict pCR after NAC by DL method based on FDG-PET/CT image. The author concluded that the DL method can predict breast cancer treatment.

The abstract is poor, please rewrite more detail to be able to understand the contents of this study.

The weak point of this study was

1. examined by a small number of cohorts with large variation.

2. no case was used for validation in the DL algorism (DLA)

The author stated that “ the study was conducted with rather a small number of patients. Future prospective and large database studies should be used to further validate and investigate our results.”

The state indicated that the algorism created by the author is not assured the reliability and reproductivity. The author should add analysis with each index (SUVmax, TLG, MTV, T and N grade,,,,) which have been assessed for the prediction of pCR in previous studies, and compared the result with the performance obtained by the DLA.

3. Unknown indication for performing surgery after NAC

4. Unknown the target lesion or residual lesion for the assessment of the effect of NAC.

I think the author analyzed the FDG-PET/CT image performed before the indication of NAC (please state it clearly). The author selected baseline FDG-PET/CT images in the cohort that performed surgical resection after NAC. How can this method be used in a clinical situation? If DLA can suggest the possibility of pCR after the NAC, it does not assure the indication of operation after the NAC.

Following the study condition, DLA may predict the pCR only in the case having an indication of surgical operation after NAC, but it can be definitely clear based on the pathological diagnosis soon after. It is not clear the significance of the result of this study toward the clinical situation.

Others

Please explain by what modality the author confirmed the radiological T and N grade, tumor size and LN lymph node level. The author stated that the inclusion criteria for the study were the clinical status of II to III according to the 8th Edition tumor-node-metastasis (TNM) classification of the American Joint Committee on Cancer Staging. Therefore, the cancer staging should be decided with consistency.

Please explain the DLA used for the detection of the primary breast tumor. Did the author validate the image picked up by DLA as correct?

Did the author measure the SUVmax of the primary breast tumor by DICOM format image before converting it to JPEG? It seems confusing from the author's description.

Digital Imaging and Communication in Medicine should be DICOM as an abbreviation.

Please explain why the author selected 5-fold for cross-validation. Considering the number of cohorts, 10 times or leave one out cross-validation is generally adopted.

Reviewer #2: Prediction of pathological complete response to neoadjuvant chemotherapy in locally advanced breast cancer by using a deep learning model with 18 F-FDG PET/CT

The presented manuscript evaluates if data derived from 18F-FDG PET/CT implemented in a deep learning algorithm are predictive for pathological complete response (pCR) after neoadjuvant chemotherapy (NAC) in locally advanced breast cancer (LABC). The authors evaluate an interesting thesis because histopathologic workup is needed by now to clarify pCR after NAC. A deep learning algorithm that uses PET/CT data to identify patients who achieve a pathologic complete response after NAC could have a therapeutic impact.

The English language is acceptable but needed further proof reading. The repetition of same sentence beginnings, especially in the introduction part of the manuscript, disturb the flow of reading for the reader. In addition, it should be standardized in the manuscript whether a space comes after/before mathematical characters. Generally, there is a space before/after a mathematical character except for percent signs.

There are further limitations in the manuscript:

- Introduction part: “Breast cancer is the most common form of cancer and the second most common cause of cancer death amongst women.”

The author is asked to provide a suitable citation.

- The author is asked to indicate in the material and methods part who has performed the data analysis and what experience exists.

- 2.3: Please give a citation for the definition of pCR.

- 2.4.: SUVmax values are measured by using ROI´s in the data evaluation. This could lead to a missing of the true SUVmax value. However, implementing image morphological markers such as tumor size, SUVmax, HU might improve the deep learning process.

- The major limitation is the reduced number of participants included in this study. Does every patient receive the same number of PET/CT examinations to avoid bias?

6. PLOS authors have the option to publish the peer review history of their article (what does this mean?). If published, this will include your full peer review and any attached files.

Reviewer #1: No

Reviewer #2: No

---

## [Author Response · Author response to Decision Letter 0]

3 Apr 2023

We have added all requested revisions and submitted in accordance with the format. 

All patients had one PET/CT imaging in the staging of breast cancer before NAC." and "Thirty one PET/CT imagings of 31 patients were used for deep learning process." PCR was defined as in the literature and all patients were divided into pathological complete response and non-pathological response. It was predicted the pathological complete response with the pet/ct images at the time of diagnosis. 

We look forward to your help and support.

---

## [Decision Letter · Decision Letter 1]

11 Aug 2023

Prediction of pathological complete response to neoadjuvant chemotherapy in locally advanced breast cancer by using a deep learning model with 18 F-FDG PET/CT

PONE-D-22-31819R1

Dear Dr. Bulut,

We’re pleased to inform you that your manuscript has been judged scientifically suitable for publication and will be formally accepted for publication once it meets all outstanding technical requirements.

Kind regards,

Erdal Karavaş, M.D.

Academic Editor

PLOS ONE

Additional Editor Comments (optional):

Dear Author,

The manuscript has been accepted as a result of the revision.

Kind regards.

Reviewers' comments:

Reviewer's Responses to Questions

**Comments to the Author**

1. If the authors have adequately addressed your comments raised in a previous round of review and you feel that this manuscript is now acceptable for publication, you may indicate that here to bypass the “Comments to the Author” section, enter your conflict of interest statement in the “Confidential to Editor” section, and submit your "Accept" recommendation.

Reviewer #1: All comments have been addressed

Reviewer #3: All comments have been addressed

2. Is the manuscript technically sound, and do the data support the conclusions?

Reviewer #1: Yes

Reviewer #3: Yes

3. Has the statistical analysis been performed appropriately and rigorously? 

Reviewer #1: Yes

Reviewer #3: Yes

4. Have the authors made all data underlying the findings in their manuscript fully available?

Reviewer #1: Yes

Reviewer #3: Yes

5. Is the manuscript presented in an intelligible fashion and written in standard English?

Reviewer #1: Yes

Reviewer #3: Yes

6. Review Comments to the Author

Reviewer #1: (No Response)

Reviewer #3: The recommendations are satisfyingly performed. The final version of the artile is sound and can be accepted.

7. PLOS authors have the option to publish the peer review history of their article (what does this mean?). If published, this will include your full peer review and any attached files.

Reviewer #1: No

Reviewer #3: **Yes: **SONAY AYDIN

---

## [Editor Report · Acceptance letter]

6 Sep 2023

PONE-D-22-31819R1 

Prediction of pathological complete response to neoadjuvant chemotherapy in locally advanced breast cancer by using a deep learning model with 18F-FDG PET/CT 

Dear Dr. Bulut:

I'm pleased to inform you that your manuscript has been deemed suitable for publication in PLOS ONE. Congratulations! Your manuscript is now with our production department. 

Kind regards, 

on behalf of

Dr. Erdal Karavaş 

Academic Editor

PLOS ONE